# Necessities in the Processing of DNA Double Strand Breaks and Their Effects on Genomic Instability and Cancer

**DOI:** 10.3390/cancers11111671

**Published:** 2019-10-28

**Authors:** George Iliakis, Emil Mladenov, Veronika Mladenova

**Affiliations:** Institute of Medical Radiation Biology, University of Duisburg-Essen Medical School, 45122 Essen, Germany; emil.mladenov@uk-essen.de (E.M.); veronika.mladenova@uk-essen.de (V.M.)

**Keywords:** DNA repair, double strand breaks (DSBs), homologous recombination repair (HRR), gene conversion (GC), c-NHEJ, alt-EJ, single strand annealing (SSA), ionizing radiation (IR), complex DSBs, high LET radiation

## Abstract

Double strand breaks (DSBs) are induced in the DNA following exposure of cells to ionizing radiation (IR) and are highly consequential for genome integrity, requiring highly specialized modes of processing. Erroneous processing of DSBs is a cause of cell death or its transformation to a cancer cell. Four mechanistically distinct pathways have evolved in cells of higher eukaryotes to process DSBs, providing thus multiple options for the damaged cells. The homologous recombination repair (HRR) dependent subway of gene conversion (GC) removes IR-induced DSBs from the genome in an error-free manner. Classical non-homologous end joining (c-NHEJ) removes DSBs with very high speed but is unable to restore the sequence at the generated junction and can catalyze the formation of translocations. Alternative end-joining (alt-EJ) operates on similar principles as c-NHEJ but is slower and more error-prone regarding both sequence preservation and translocation formation. Finally, single strand annealing (SSA) is associated with large deletions and may also form translocations. Thus, the four pathways available for the processing of DSBs are not alternative options producing equivalent outcomes. We discuss the rationale for the evolution of pathways with such divergent properties and fidelities and outline the logic and necessities that govern their engagement. We reason that cells are not free to choose one specific pathway for the processing of a DSB but rather that they engage a pathway by applying the logic of highest fidelity selection, adapted to necessities imposed by the character of the DSB being processed. We introduce DSB clusters as a particularly consequential form of chromatin breakage and review findings suggesting that this form of damage underpins the increased efficacy of high linear energy transfer (LET) radiation modalities. The concepts developed have implications for the protection of humans from radon-induced cancer, as well as the treatment of cancer with radiations of high LET.

## 1. Introduction: General Considerations for DSB Repair Pathway Utilization

Cancer is intricately linked to the formation of chromosomal translocations, one of the most prominent manifestations of genomic instability that underpin the evolution of this disease [1,2,3]. Chromosomal translocations are the direct consequence of erroneous processing of DNA double strand breaks (DSBs) generated in cells as a consequence of problems arising during DNA replication, or as programmed events in meiosis, or during the development and maturation of the immune system. DSBs also arise, directly or indirectly, as a consequence of cell exposure to various chemical and physical agents that damage the DNA.

It is widely recognized that in vertebrates, DSBs represent a particularly dangerous DNA lesion because they directly disrupt the integrity of the DNA molecule and generate a condition where fundamental concepts evolved to repair other forms of DNA lesions fail [4,5,6,7]. This is because DSBs affect both DNA strands and compromise the key principle utilized to repair lesions confined to one strand: The use as a template of the complementary, undamaged strand to restore sequence and integrity in the damaged strand [8].

This characteristic of DSBs imposes the expectation that their processing will be problematic and therefore inefficient and slow. Strikingly, however, work carried out in the last 30 years documents an impressive DSB-processing system with an extraordinary total repair capacity, comprising multiple repair pathways, some of which operate with very high speed [4,5,6,7]. This DSB-processing system ensures that the vast majority of DSBs induced in cells by physical agents, such as ionizing radiation (IR), but also by diverse chemicals, are often processed faster than DNA lesions confined to one DNA strand.

Despite these apparently impressive characteristics of the DSB-processing system, DSB induction remains a high-risk event as amply documented by the long-described detrimental effects of IR: Point mutations, as well as genomic rearrangements, causing cell death and most prominently cancer [9]. As a consequence of this inherent severity of the DSB as DNA lesion, a highly complex network of cellular responses evolved that is promptly activated when cells detect DSBs in their genome. This so-called “DNA damage response (DDR)” [10] originates exclusively from DSBs (and single-stranded DNA regions), regulates processes that modify nearly every metabolic activity of the cell, and sets the stage for processing, adaptation, or programmed cell death. 

A highly consequential recognition from recent in-depth analysis of DDR and DSB repair is that the efficient overall DSB processing mentioned above is achieved by the coordination of multiple pathways, which have very distinct operational requirements and which surprisingly produce results that are not equivalent [4,5,6,7]. This is quite unique for DSB processing, as alternative pathways normally execute a biological process in a more or less equivalent manner and reach the same final result in a different way. 

For DSB processing, in contrast, while all characterized pathways remove the physical rupture from the DNA, they do so with widely different accuracy in terms of restoring the original sequence in the vicinity of the DSB, and occasionally they even join the wrong DNA ends, causing translocations. 

Four different pathways evolved to process DSBs. The homologous recombination (HR) repair-based pathway, gene conversion (GC), is the only error-free pathway capable of restoring both the DNA integrity and the original DNA sequence. Classical non-homologous end joining (c-NHEJ), alternative end joining (alt-EJ), and single-strand annealing (SSA), another HR-dependent pathway, remove the break from the genome, but can cause mutations, deletions, and translocations owing to functional characteristics that are inherent to their mechanistic base and which are described in the following section [4,5,6,7]. As a consequence, these pathways are not alternative ways of processing a well-defined DNA-lesion, achieving a well-defined goal: The full restoration of the genome. Rather, as we will see below, their evolution suggests that they serve to handle different forms of DNA ruptures; accepting reduced outcome precision to achieve DSB removal and thus first-line genomic integrity, when difficulties arise in DSB processing. 

A working hypothesis guiding our research activities is that a key parameter in DSB repair pathway engagement is the degree of chromatin destabilization generated in the genome by a DSB. We postulate that this chromatin destabilization affects each of the DSB processing pathways in a different manner.

Several parameters that may determine the degree of chromatin destabilization generated by a DSB have been identified and are presently under intensive investigation to establish their contribution to DSB processing. Thus, a DSB generated in euchromatin is likely to have different consequences than a DSB generated in heterochromatin. Even within euchromatin, a DSB generated in an actively transcribed region of the genome is likely to be highly destabilizing. Also, DSBs generated in regions of active DNA replication will generate pronounced chromatin destabilization and possibly also a disruption of the replication fork. 

Furthermore, chromatin is highly dynamic in each phase of the cell cycle and can undergo dramatic changes when cells transit from one phase of the cell cycle to the next. Perhaps the most dramatic such changes in chromatin organization occur when cells enter mitosis, as well as after completion of mitosis when cells enter G_1_ phase. DSBs induced before or during mitosis are therefore likely to generate severe chromatin destabilization, as amply documented by the observation that mitosis is by far the most radiosensitive phase of the cell cycle [11]. 

In the case of IR, the form of DSBs generated adds another level of complexity and a whole new set of parameters to the above-described aspects of chromatin organization and activities [4]. In stark contrast to DSBs induced by restriction endonucleases, IR generates different forms of DSBs with a widely differing potential for chromatin destabilization. This is because IR generates DSBs through clusters of ionizations that hit and break both DNA strands. An ionization cluster may contain two ionizations but may also contain many more (Figure 1). Depending on this number of ionizations, the DNA may not only be broken but additionally be damaged at multiple locations in the vicinity of the break generating a so-called “complex” lesion (see below). Such accumulation of additional DNA damages in the vicinity of the DSB is likely to increase the local destabilization of chromatin as a consequence of the parallel recruitment to the DSB site of proteins processing base damages and single strand breaks. Additional levels of lesion complexity have been described and will be discussed in detail in the following sections. 

We postulate that the degree of chromatin destabilization by a DSB will be determined by the above outlined physical and biological parameters and that the degree of chromatin destabilization will, in-turn, determine the pathway that will engage in the processing of this DSB. All available information suggests that the prime goal of this integrated processing system is to leave no DSB unprocessed and that this feat is achieved by joining together, even genomically incongruent, DNA ends. Although this goal is achieved at the cost of events that change and destabilize the entire genome, such as DNA sequence alterations, nucleotide losses, and translocations, such an outcome is clearly preferable to the broken genome that would be left behind when high fidelity processing mechanisms fail in the chromatin environment of the DSB. 

Along this line of reasoning, unrepaired DSBs, or DNA ends detected after completion of bulk DSB processing, reflect a failure to bring two DNA ends in proximity to each other and join them together. This concept also explains why higher eukaryotes rely heavily on pathways that simply join DNA ends together independently of their origin (c-NHEJ, alt-EJ) and less so, at least apparently, on more complex but error-free processes, such as GC—as yeast and bacteria actually do! 

Indeed, end-joining pathways may primarily and more appropriately be considered as DNA-end management/scavenging systems rather than genuine repair mechanisms, designed to remove DNA ends from the genome by bringing them together and joining them to each other. They may actually reflect late events in evolution that contribute to the management of integrity of increasing-size genomes. Fatal events causing cell death and cancer may thus be accidents that are “taken into account” during engagement of a particular pathway for the processing of a specific DSB. 

Considering all the above information, we propose that the engagement of a particular DSB processing pathway is dictated by necessity rather than by choice. Our working hypothesis is that cells must in principle be evolutionarily programmed to use first the DSB processing pathway with the highest probability of restoring an intact genome, but that they will also adaptively recruit lower fidelity options if the state of chromatin at the rupture site generates conditions that compromise this pathway. The concept of necessity suggests a hierarchy in DSB processing, with the highest fidelity pathway interrogated first. Pathways of decreasing fidelity will be interrogated, sequentially, only when the previous higher fidelity pathway fails in the context of existing chromatin organization and activity. The “last resort” pathways will put together any DNA ends including those belonging to different chromatin regions and even to different chromosomes. 

Such hierarchical interrogation of existing DSB processing pathways in the removal of DSBs is hypothesized to reflect an inherent logic in their evolution aiming at maximizing the preservation of genetic information after adaptation to realities. It also implies that the cell is not “choosing” among one of the four repair pathways, but rather that it applies the above “logic” to every DSB and always ends up using the best pathway under the circumstances.

Below, we summarize key characteristics of the DSB processing pathways to illustrate the basis for the above concepts and review results of experiments that describe how these concepts and hypotheses can be applied to understand the role of DSB complexity in the engagement of a processing pathway. We will illustrate how radiation modalities of increasing LET induce DSBs of increasing complexity and will show that increased DSB complexity compromises DSB processing by c-NHEJ for an as of yet undefined subset of DSBs. Finally, we will use this property to validate in defined biological models a type of complexity, DSB-clusters, as one of the key lesions underpinning the increased efficacy of high LET radiation.

## 2. Characteristics of Pathways Processing DSBs

### 2.1. General Considerations

As outlined above, the most intriguing characteristic of the DSB processing apparatus is that it integrates four pathways, each of which processes DSBs using different molecular mechanisms (Figure 2). Seen under the prism of the above developed rationale, this mechanistic plurality endows cells with options to deal with DSBs generating in chromatin different degrees of destabilization. As a result, the four pathways do not reflect alternative options that can be freely selected on stochastic grounds to handle consequence-equivalent DSBs. Therefore, we will refrain from using the term “DSB repair pathway choice” to avoid the impression that the cell is free to engage any of the four pathways for all DSBs independently of form, localization in chromatin, or ongoing activities in chromatin. We will attempt to show that there is actually only a limited degree of freedom in pathway selection during the processing of any given DSB. We outline next the four DSB repair pathways and the consequences of their operation to genome integrity and rationalize why we postulate their hierarchical engagement based on necessities.

### 2.2. Rapid Removal of DNA Ends by c-NHEJ

Normally, DSBs generated by IR have blunt ends or contain very short protruding single stranded DNA ends [4]. C-NHEJ is highly efficient in removing such DNA ends from the genome by joining them together, de-facto removing in this way the DSB. C-NHEJ starts with the extremely rapid and impressively strong binding of KU70/KU80 heterodimer (KU) to DNA ends that mediates the immediate recruitment of the large DNA-dependent protein kinase catalytic subunit (DNA-PKcs) to form the active DNA-PK holoenzyme [4,5,6,7,12,13] (Figure 2). Owing to its very large size, the DNA-PK holoenzyme is thought to function as a recruitment scaffold for additional components of c-NHEJ, and that its kinase activity regulates the process by phosphorylating several proteins, including itself.

Before joining by the LIG4/XRCC4/XLF/PAXX complex, each of the DNA ends needs to be “cleaned” to enable ligation. This is because IR breaks DNA by damaging the sugar moiety of terminal nucleotides, remnants of which need to be removed. This processing utilizes Artemis, Aprataxin and Aprataxin, and PNK-like factor (APLF). DNA synthesis by members of the X family DNA polymerases, Pol λ and Pol µ, fills gaps in the DNA and adds new nucleotides in a non-complementary manner, ultimately completing the restoration of the DNA molecule [12,14]. 

The c-NHEJ molecular apparatus functions highly efficiently; assisted by the simplicity of the catalyzed reaction and the high numbers of KU and DNA-PKcs molecules in the cell, it removes DNA ends from the genome with unparalleled speed in a cell cycle-independent manner [4]. Indeed, c-NHEJ operates faster than many of the repair pathways processing base lesions confined to one DNA strand [4]. Yet, c-NHEJ joins any DNA ends that happen to be close by, i.e., it has no built-in mechanisms ensuring the joining of the original DNA ends. As a result, c-NHEJ can catalyze the formation of translocations [15,16]. However, its extraordinary speed and the fact that upon DSB generation by IR the two cognate DNA ends are very close to each other in space, c-NHEJ will very frequently join together the “correct” ends, thus suppressing translocation formation. It follows that translocation formation catalyzed by c-NHEJ likely involves DNA ends that drifted apart to rejoin with unrelated, similarly drifted-apart DNA ends. Moreover, and as obvious from its molecular setup, c-NHEJ has no built-in mechanisms to restore the DNA sequence at the DSB junction when cognate ends are rejoined, and frequently, additions or deletions of nucleotides are observed (Figure 2). However, c-NHEJ-mediated alterations in the DNA sequence are small and involve deletions or additions of only a few nucleotides [4,5,6,7,12,13].

In conclusion, c-NHEJ features as a surveillance mechanism, swiftly scavenging DNA ends from the genome by joining them pairwise as soon as they happen to be close to each other in space. It thus keeps the genome free of DNA ends, which improves its stability. A conundrum generated by the above outlined molecular characteristics of c-NHEJ is how the three remaining DSB processing pathways engage the DSB in a c-NHEJ-proficient background and why the cell apparently favors a fundamentally error-prone pathway. Below, we present forms of DSBs that compromise c-NHEJ and facilitate the utilization of the remaining DSB processing pathways.

### 2.3. DNA-End-Resection-Dependent Rejoining of DSBs by GC, SSA, and Alt-EJ

The ends of DSBs that are not removed from the genome by c-NHEJ are prepared for processing by three alternative pathways all of which begin, albeit to a different extent, with the exonucleolytic degradation of the 5′-strands of each DNA end to generate 3′-single-stranded DNA (ssDNA), in a reaction known as DNA-end resection (Figure 2). DNA-end resection serves a dual purpose: First, it helps free-up the DNA ends from KU, which has very low affinity for ssDNA; and second, it initiates the alternative modes of DSB processing described next [13,17]. DNA-end resection utilizes among others the activities of the MRE11/RAD50/NBS1 (MRN) complex (), as well as those of CtBP-Interacting Protein (CtIP), Exonuclease 1 (EXO1), DNA replication helicase/nuclease 2 (DNA2), and the Bloom’s syndrome (BLM) helicase, and generates ssDNA with 3′-overhangs that is stabilized by Replication Protein A (RPA). This structure subsequently initiates GC, SSA, or alt-EJ. Resection is strongly regulated throughout the cell cycle at multiple levels and this generates a strong cell cycle dependence for all resection-dependent pathways of DSB processing [18].

#### 2.3.1. Gene Conversion (GC)

The most salient feature of GC is that it requires a homologous template in the form of the sister chromatid [19,20,21,22]. During GC, the above described DNA-end resection reaction proceeds to remove a few thousand base pairs from both DNA ends to generate equally long 3′ ssDNA strands that are stabilized by RPA. RPA is exchanged by RAD51 in a reaction that is facilitated by the RAD51 paralogs (RAD51B, RAD51C, RAD51D, XRCC2, and XRCC3) and BRCA2 [19,20,21]. The RAD51 nucleoprotein filament subsequently invades and anneals to the homologous region in the sister chromatid to form a structure known as a Holliday junction (HJ), and the 3′ end is extended by templated DNA synthesis aided by RAD54. The spatial proximity between sister chromatids imposed shortly after completion of DNA replication by the loading of cohesins greatly facilitates the homology search by the RAD51 coated ssDNA and reduces the degrees of freedom of the DNA ends and thus their probability to drift apart. Following HJ formation and initiation of DNA synthesis, and depending upon whether the second DNA end is similarly captured to form a double HJ, the reaction can continue along different paths, a detailed description of which is beyond the scope of this review. However, in the case of IR-induced DSBs, the most likely scenario is that the HJ is resolved after extension and that the released ssDNA anneals with the similarly end-resected ssDNA of the other DNA end to reconstitute a dsDNA molecule with gaps that are filled and ultimately ligated to fully restore the DNA molecule [19,20,21]. 

The constraints of the above described mechanism ensure that exclusively, the original ends of DSB will be joined together. Furthermore, the templated DNA synthesis on the identical sister chromatid enables GC to faithfully restore DNA sequence in the vicinity of the DSB. As a result, GC operates in an error-free manner without causing translocations or sequence modifications in the cellular genome (Figure 2). On the other hand, GC can only take place post-replicatively, partly during the S-phase and fully during the G_2_-phase of the cell cycle [18]. In addition, the number of steps involved, particularly the invasion step, suggest that GC will be a relatively slow process.

#### 2.3.2. Single Strand Annealing (SSA)

SSA is also a homology dependent pathway of DSB processing [22,23,24,25]. However, in contrast to GC that relies exclusively on homology found in the sister chromatid, SSA normally utilizes homologous regions present in the same DNA molecule. During SSA, resection of DNA ends may extend even further than after GC, reaching at times a few hundred thousand base pairs depending on the distance between the flanking homologies. Notably, the abundance of repetitive sequences in the human genome amply provides homologies for the engagement of SSA. Nevertheless, systematic analysis of the level of SSA engagement in IR-induced DSB processing has only relatively recently begun to be analyzed systematically [24].

In contrast to HRR, SSA does not require RAD51 but uses instead the strand annealing protein RAD52 to anneal the DSB-flanking homologous DNA sequences, as they become exposed after DNA-end resection [22,23,24,25]. SSA is always associated with the deletion of the DNA segment delimited by the utilized regions of homology. As a consequence, it also requires the removal of the ssDNA flaps forming after annealing of the homologous regions. This function is provided by the XPF–ERCC1 and MSH2–MSH3 complexes and is followed by DNA ligation to restore integrity in the DNA molecule. SSA is highly error prone owing to the large deletions it generates and can be promiscuous in partner selection, thus forming translocations [22]. However, when SSA operates post-replicatively, cohesin-mediated stabilization is expected to reduce the drifting apart of DNA ends and thus the formation of translocations. Thus, despite the use of homology, SSA is a highly error-prone DSB processing pathway and the requirement for resection makes it cell cycle dependent with maximum activity during S- and G_2_-phase.

#### 2.3.3. Alternative End-Joining (Alt-EJ)

Alt-EJ operates on similar principles as c-NHEJ and simply joins two DNA ends together [4,12,26,27]. However, in contrast to c-NHEJ, alt-EJ often benefits from DNA-end resection that exposes microhomologies. These microhomologies are typically less than 10 base pairs long and may help in the annealing between the two resected DNA ends that facilitates their joining. This property justifies the alternative designation of this pathway as microhomology-mediated end-joining (MMEJ) [28,29]. However, it has not been formally demonstrated that all processing of IR-induced DSBs requires resected DNA ends, and it is likely that a fraction of DNA ends is processed by alt-EJ without being measurably resected.

Proteins implicated in alt-EJ, in addition to the above-mentioned resection proteins, are often not dedicated to this pathway and are involved in other aspects of the DNA metabolism as well. For example, Poly (ADP-ribose) Polymerase 1 (PARP-1) facilitates alt-EJ but also has many additional functions in the cell [4,12,26,27]. A rather dedicated newcomer protein to alt-EJ is Polθ, an A family DNA polymerase, encoded by the *POLQ* gene [30,31,32]. Polθ has a C-terminal polymerase domain, as well as an N-terminal helicase-like domain that stabilizes the annealing of two long 3′ ssDNA overhangs, even when they show little homology. In the structures generated in this way, the polymerase activity of Polθ extends the 3′ end using the annealed partner as a template [31]. The final ligation step of alt-EJ is mediated by DNA ligases I and III (LIG1 and LIG3) while the linker histone H1 may serve as an alignment factor [4]. Although alt-EJ is active throughout the cell cycle [25,26,27], it shows strong cell cycle-dependent activity fluctuations with a maximum in G_2_-phase and reduced activity in G_1_, as expected from its dependence on DNA-end resection [27]. Indeed, alt-EJ is often ablated in resting G_0_ cells [33,34,35]. 

Similarly to c-NHEJ, alt-EJ joins any DNA ends independently of origin. However, because it operates with slower kinetics than c-NHEJ and is carried out by a much less coordinated apparatus, a drifting apart of the DNA ends is more likely and as a consequence also the probability of translocation formation. Alt-EJ has also no built-in mechanisms to restore DNA sequence at the DSB junction. This fact, together with the often-required DNA-end resection, increases the frequency of deletions and other sequence alterations at the junction; indeed, these changes in the genome are much more extensive than those generated by c-NHEJ. 

Collectively, the above outline indicates that only GC operates in an error-free manner. All remaining DSB processing pathways are error prone and all can, in principle, contribute to the formation of chromosomal translocations and sequence alterations at the DSB junction. Below we present results rationalizing the evolution of repair pathways with such diversity.

## 3. IR Induces DSBs of Different Complexity

As mentioned above, DSBs form in irradiated cells as minority lesions [36,37] when ionization clusters are produced in the vicinity of the DNA, damaging the molecule at multiple locations, within a few nucleotides, simultaneously (Figure 1). IR can be in the form of photons, such as X-rays or gamma rays, or in the form of energetic particles—from electrons, protons, or neutrons to nuclei as large as uranium. A key early discovery of radiation biology is that different forms of IR generate widely different biological effects [38]. This is illustrated in Figure 3, which shows cell survival, measured by assessing the ability of cells to form colonies, as a function of radiation dose; similar responses have been confirmed for many other endpoints. It is evident that 56Fe+ ions kill M059K cells much more efficiently than X-rays. Since similar numbers of ionizations are generated at equal doses of the different radiation modalities [37,39,40,41], additional parameters must underpin the increased effectiveness of alpha particles and other high LET radiation modalities. One such parameter is LET (linear energy transfer), which describes the amount of energy imparted (and thus of ionizations induced) per unit of particle-track length as it traverses the cell (Figure 1). The higher the LET of radiation, the higher the proportion of clustered-ionization events at any given dose. 

Because increased ionization clustering increases the probability of DSB induction, it was at first hypothesized that the increased efficacy of high LET radiations derives from increased induction of DSBs. Surprisingly, however, extensive work testing this hypothesis demonstrated that per unit of dose, radiation modalities of different LET produce similar numbers of DSBs [42,43,44,45,46,47,48] (Figure 4a), although an increase in intermediate fragments has also been reported [49].

These results led to the follow-up hypothesis that high LET radiations are more efficient biologically because they generate more complex DSBs that are difficult for the cell to process [36,50,51,52,53]. Since a biologically relevant classification of DSBs based on complexity remains elusive, we made an attempt to classify DSBs based on the number of damages they contain [4] (Figure 5). In this classification, the simplest form of DSB is the one generated by restriction endonucleases (REs) [4], which break the DNA without chemically altering any of its nucleotides. IR on the other hand induces DSBs by chemically altering nucleotides: In their simplest form, IR-induced DSBs break the DNA by damaging the sugar of the sugar–phosphate backbone and frequently generate a 3′-damaged sugar in the form of phosphoglycolate and a 5′-OH that must be removed prior to end joining [54,55] (see above) (Figure 5). Therefore, these DSBs are considered more complex than those induced by REs. A further increase in complexity is generated when additional forms of damages, such as base damage or single strand breaks, are present within one helical turn of the DSB [56]. It is this form of complexity that has been found to increase with increasing LET and is therefore frequently invoked to explain the enhanced biological effects. The hypothesis is that in this form of DSBs, processing is hampered by such non-DSB lesions that generate obstacles in the function of DSB-processing pathways, and which in addition destabilize chromatin at the site. 

Another form of complexity that has been frequently considered to explain the increased efficacy of high LET radiation is that of DSB clusters (Figure 5). DSB clusters are likely to be highly destabilizing for chromatin and this destabilization, in conjunction with chromatin location and chromatin activities, may seriously compromise their faithful restoration [4]. Mathematical modeling shows that the physical characteristics of high LET IR increase the probability of induction of DSB clusters, thus explaining the increased efficacy of high LET IR [4,56].

Notably, for both forms of DSB complexity, the supportive evidence for their relevance to high LET effects derives largely from mathematical modeling. For DSB complexity owing to the presence of non-DSB lesions at the DSB site, biochemical experimental evidence exists but is indirect. As a result, the relative contribution of the two forms of complexity in the increased effectiveness of high LET radiation remains unknown. 

Below, we will describe a biological model system we recently developed to generate in cells simple DSBs and DSB clusters using the I-SceI homing meganuclease [57]. The system is based on the genomic integration as multiple copies of constructs harboring different configurations of the I-SceI recognition-sequence, such that upon expression of the enzyme, single DSBs or DSB clusters of different complexity are generated and their processing is followed at different endpoints. Since validation of this model system requires information on the processing of DSBs generated by high LET IR, we review next the available relevant information.

## 4. A Subset of DSBs Induced by High LET IR Compromises c-NHEJ

Despite increased complexity, DSBs induced by high LET IR, such as 56Fe+ ions, are actively processed, albeit with a somewhat reduced efficiency compared to DSBs induced by low LET IR, such as x-rays [58] (Figure 4b). Similar results have been reported in other cell lines and for additional forms of high LET IR. DSB processing following high LET radiation exposure markedly depends on c-NHEJ, as mutants with defects in this pathway show clear reduction in the rejoining efficiency (Figure 4b). Residual DSB processing at these high doses reflects the function of alt-EJ and possibly also of SSA. It can therefore be concluded that a significant proportion of high-LET-IR-induced DSBs are processed by c-NHEJ [58].

A striking observation with high LET IR is that when cell survival is analyzed in c-NHEJ mutants, the LET-dependent enhancement of cell killing disappears [58]. This is illustrated in Figure 3a, which shows that M059J cells defective in DNA-PKcs are equally sensitive to 56Fe+ ions and x-rays. Similar observations have been reported by others in a variety of c-NHEJ mutants [59,60,61,62,63,64]. Notably, GC mutants are markedly more sensitive to high as compared to low LET IR [62], suggesting that something special happens with c-NHEJ (Figure 3b). 

A possible interpretation for this interesting observation is that wild-type cells gain their radioresistance to low LET IR, as compared to high LET IR, by repairing, using c-NHEJ, a subset of DSBs, which after exposure to high LET IR change form and become refractory to processing by c-NHEJ. As a consequence, and because high and low LET IR have similar yields of DSBs (Figure 4), c-NHEJ-deficient cells are equally sensitive to high and low LET IR. This notable observation is in our view an indication that the form of DSBs induced by a particular radiation modality can at times compromise DSB-processing pathways and that c-NHEJ is particularly susceptible to changes in DSB form generated by increases in the LET of the radiation used. However, why should c-NHEJ be more susceptible to forms of DSBs generated by high LET IR than other DSB repair pathways and what kind of change in DSB form actually suppresses c-NHEJ?

We hypothesized that the mega-molecular apparatus of c-NHEJ, consisting of proteins the size of DNA-PK and coordinated by several additional interacting components, must operate precisely, like a “Swiss watch”, to remove DSBs with the unparalleled speed measured experimentally. This overall operation accuracy and speed are likely to be particularly sensitive to the generation of DSB clusters. We postulated that, particularly, DSB clusters in which the individual DSBs are located close together are likely to destabilize its function. In a cell exposed to IR, only a small subset of DSBs (chromatin rupture events more precisely) will be caused by a cluster of DSBs. However, their yields will increase and the distance between the individual DSBs in a cluster will decrease with increasing LET of radiation.

It is not possible to test the hypothesis of DSB cluster formation as a key cause, both for the high LET radiation action as well as for the associated suppression of c-NHEJ, by exposing cells to IR. This is because IR induces a daunting spectrum of DNA lesions distributed throughout chromatin at locations that are different for each cell, thus making molecular analysis of their consequences impossible. To address this problem, we engaged in the development of a molecularly defined biological model of DSB clusters in order to examine their suppressive consequences on c-NHEJ and analyze their processing by GC, alt-EJ, and possibly also SSA. Below, we briefly describe this model and summarize results confirming that complex DSB clusters suppress the function of c-NHEJ while allowing processing by the remaining repair pathways. 

A relevant observation worth mentioning in the context of the present section is the generation of DSBs in cells exposed to protons, a low LET radiation modality, that are preferably processed by GC. Specifically, it was observed that cells lacking proteins of the homologous recombination DSB repair pathway are more sensitive to proton than to photon irradiation. The chromatin context and the specifics of DSB induction that generate this dependence on GC remains to be elucidated. However, the observation is relevant for the application of protons in the treatment of cancer as they may be particularly indicated for cancers that share molecular features of BRCA-mutant tumors, “BRCAness” [65,66].

## 5. DSB Clusters Suppress c-NHEJ and Cause Translocations

A RE frequently used to induce DSBs in the human genome and study its processing requirements is the I-SceI homing endonuclease. I-SceI recognizes an 18 base-pair sequence that is not present in the vertebrate genome (I-SceI is an intron-encoded endonuclease present in the mitochondria of *Saccharomyces cerevisiae*), but which can be introduced, typically as a single copy, in the form of a construct designed according to the question addressed [67,68,69,70]. The integration site can be random or pre-selected with the construct targeted to that particular location. The generated clonal cell lines harbor the construct at a genomic location that can be precisely known and DSBs are induced by transfecting cells with vectors expressing the I-SceI endonuclease (Figure 6) [35,71].

Although I-SceI, like all REs, induces the simplest form of DSBs possible (see above), it has been successfully used to elucidate key requirements of DSB-processing pathways. We opted therefore for this system and designed constructs allowing study of the processing of DSB clusters by available DSB-repair pathways, focusing first on the elucidation of the above described unique characteristics of high LET radiation. 

For this purpose and in deviation from available I-SceI cell systems, we generated CHO cell lines containing multiple I-SceI integration sites (rather than the typical single integration) to mimic the multiple DSBs typically induced in irradiated cells. Furthermore, we designed constructs harboring the I-SceI site either as a single copy to generate single DSBs, or as pairs and quadruplets to generate DSB clusters. The distances between the DSBs vary between approximately 50 and 200 (average nucleosome distance) base pairs. Figure 6 summarizes the cell lines used in the study, the type of DSB cluster generated by restriction with I-SceI, and the orientation used for the I-SceI recognition sequences in order to generate ligatable (D) or unligatable (R) apical ends. Only DSBs are generated in cells in this model system. SSBs and base damages, which far outnumber DSBs in cells exposed to IR [4], are completely absent.

This model system is designed on the assumption that transient expression of I-SceI will rupture chromatin by generating either single DSBs or DSB clusters. Each such event is considered as a single chromatin rupture of “complexity” proportional to the number of DSBs involved. Indeed, single DSBs and DSB clusters initiate qualitatively and quantitatively similar initial γ-H2AX foci formation at numbers equivalent to the integration sites, although 53BP1 accretion appears enhanced in regions of chromatin-harboring DSB clusters. Chromatin restitution may occur by ligation of proximal or apical DNA ends [57].

The first test of DSB clusters as a putative cause for the increased efficacy of high LET IR is summarized in Figure 6c, which shows colony formation in the different clones after expression of I-SceI. It is evident that as with increasing LET, increased killing is observed with increasing cluster complexity. Indeed, the vast majority of cells harboring I-SceI-quadruplets succumb to this form of DNA damage. Thus, DSB clusters are more likely to confer lethality than single DSBs, with cell lethality increasing with the increasing number of DSBs per cluster, or when apical ends are unligatable.

Defects in c-NHEJ are associated with an increase in the formation of translocations. Analysis of translocations forming during processing of single DSBs and DSB clusters is shown in Figure 6d and demonstrates the highest incidence after induction of DSB clusters. Notably, inhibition of c-NHEJ with an inhibitor of DNA-PKcs increases the formation of translocations following the generation of single DSBs and DSBs pairs but fails to have an effect following the generation of DSB quadruplets. Since cell survival and translocation formation correlate, this result recapitulates the lack on enhanced killing with increasing LET in c-NHEJ-deficient cells. We consider this observation therefore as corroborating our hypothesis that DSB clusters compromise c-NHEJ. The strong reduction in the frequency of translocation formation from DSB quadruplets in cells incubated with an inhibitor of PARP-1, (PJ34) suggests that alt-EJ is functional under these conditions. A possible role of SSA is presently under investigation, as is also the in-principle activation of GC.

## 6. Conclusions

We developed a rationale for the evolution in vertebrate cells of four mechanistically distinct pathways for the processing of DSBs, which surprisingly have very different capacities to reconstitute the original genome. We reasoned that in cells, these pathways must be wired in ways allowing first the engagement of the highest fidelity pathway—following an inherent logic aiming at maximizing genomic integrity. We further argued that when cells use a pathway of lower fidelity to process a specific DSB, they do so to address necessities that arise from the suppression of a high-fidelity pathway during processing in the context of chromatin. Such necessities arise as a consequence of the DSB location in chromatin, as well as a consequence of chromatin activities in the affected chromatin domain that may interfere with the function of a repair pathway. Similar necessities also arise as a consequence of increasing DSB complexity. Results obtained using a defined biological model system demonstrate that DSB clusters compromise c-NHEJ. By comparing the responses to high-LET IR of cells deficient in c-NHEJ, we developed the hypothesis that a significant proportion of the increased killing generated by high-LET IR derives from the generation of DSB clusters. The results support a model according to which cells are not freely choosing one pathway for the processing of a DSB, but rather apply logic, adapted to necessities generated by realities deriving from the character of the DSB being processed. 

Among the four processing pathways available to vertebrate cells for the removal of DSBs, only GC faithfully restores the genome to its original state. All remaining pathways, but c-NHEJ in particular, operate as DNA-end scavenging mechanisms, ultimately aiming at leaving no DNA ends open in the genome. The extremely high efficiency of c-NHEJ in higher eukaryotes begs the question regarding the benefits of its evolution. The large differences in the karyotypes between otherwise genetically related species and the newly discovered phenomenon of chromothripsis in cancer support the hypothesis that c-NHEJ may function as an accelerator of evolution.

## Figures and Tables

**Figure 1 cancers-11-01671-f001:**
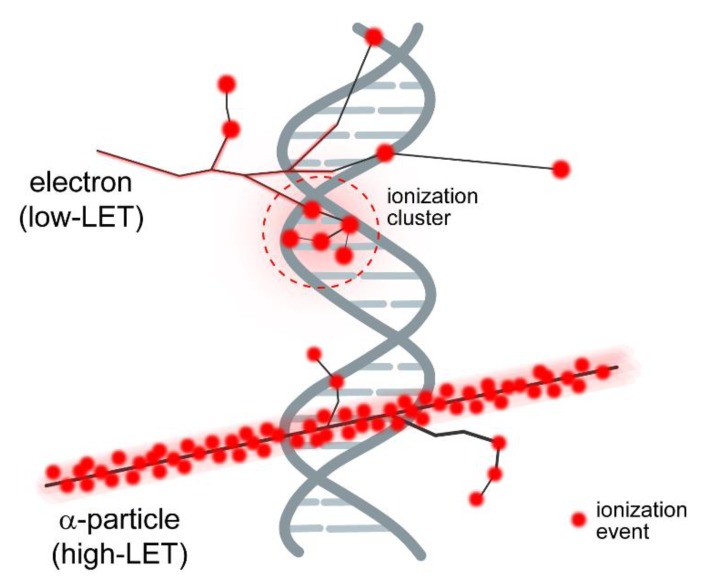
Distribution of ionizations generated in DNA-containing space by electrons of low LET, or high LET alpha particles. Note the increased ionization density from the high LET particle. When ionization events simultaneously break both DNA strands, DSBs are generated. The schematic considers for simplicity only direct IR effects, i.e., ionizations occurring directly on the DNA. However, damages generated in the DNA indirectly (indirect effects) are also very important for the overall effects of IR, i.e., by radicals produced when water molecules in the vicinity of the DNA are ionized. Such events add to the complexity of events inducing DSBs but do not fundamentally alter the rationale developed here. The contribution of indirect effects to the overall effect drops with increasing LET of the radiation used.

**Figure 2 cancers-11-01671-f002:**
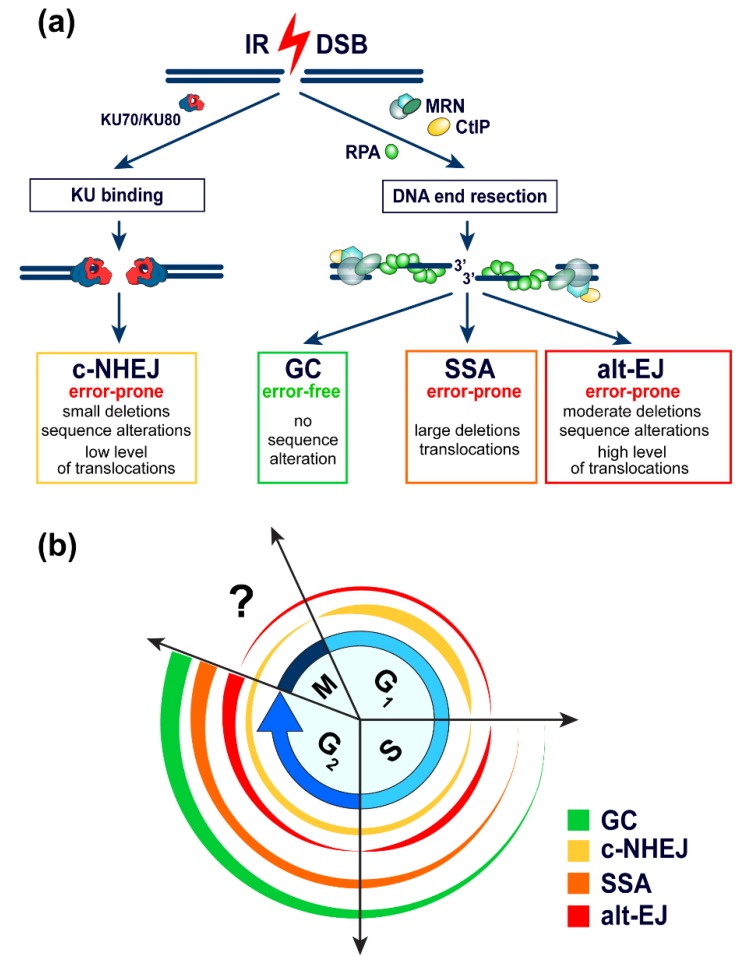
(**a**) Repair of DSBs by DNA end resection independent and DNA end resection dependent DSB repair pathways. Classical non-homologous end joining (c-NHEJ) is initiated by the strong binding of KU heterodimer and the recruitment of DNA-PKcs protein kinase, which is believed to block and attenuate DNA end resection dependent DSB processing. Gene conversion (GC), single-strand annealing (SSA), and alternative end joining (alt-EJ) are initiated by precisely controlled DNA end resection. Only GC results in faithful restoration of the DNA sequence around the DSB. All remaining DSB repair pathways are error-prone and are characterized by the generation of genomic alterations. The possible ramifications of the distinct characteristics of the different DSB repair pathways and the significance of their error-prone nature are discussed in the text. (**b**) Outline of the cell cycle dependence of the above DSB processing pathways.

**Figure 3 cancers-11-01671-f003:**
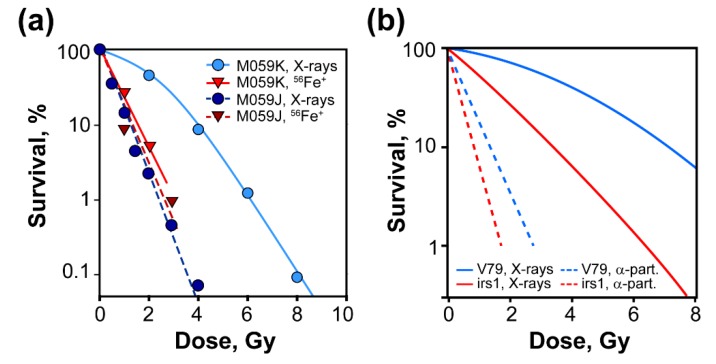
(**a**) Representative survival experiments with DNA-PKcs-proficient (M059K) and DNA-PKcs-deficient (M059J) cells exposed to low LET (x-rays) and high LET (56Fe+ ions, LET = 151 keV/µm). Note that c-NHEJ deficiency associated with the DNA-PKcs defect eliminates the LET-dependent radiosensitization observed with wild-type cells. The results depicted here are re-drawn from Singh et al. to illustrate the concepts developed in the present review. (**b**) Similar results obtained with the XRCC2-deficient irs1 mutant of V79 that generates a strong defect in GC. The results shown have been traced from Hill et al.

**Figure 4 cancers-11-01671-f004:**
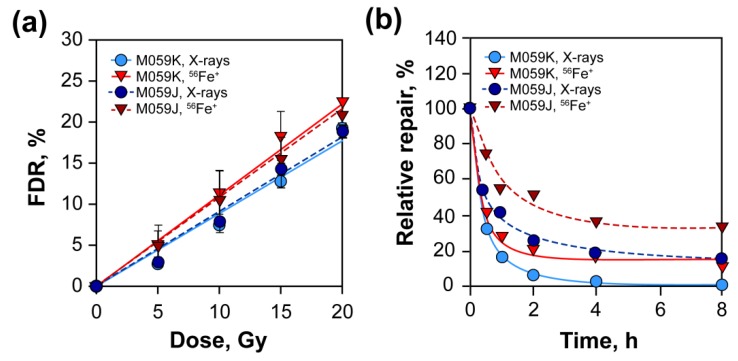
Pulsed-field gel electrophoresis (PFGE) experiments of c-NHEJ-proficient (M059K) and c-NHEJ-deficient (M059J) cells exposed to x-rays and 56Fe+ ions (LET 151 keV/µm). (**a**) Dose response curves of M059K and M059J cells exposed to low or high LET IR. Note that the induction of DSBs is not affected by the LET or the defect in c-NHEJ. (**b**) Repair kinetics of M059K and M059J cells exposed to low or high LET IR. Note that despite subtle differences, c-NHEJ-proficient and -deficient cells actively repair DSBs after exposure to either high or low LET IR.

**Figure 5 cancers-11-01671-f005:**
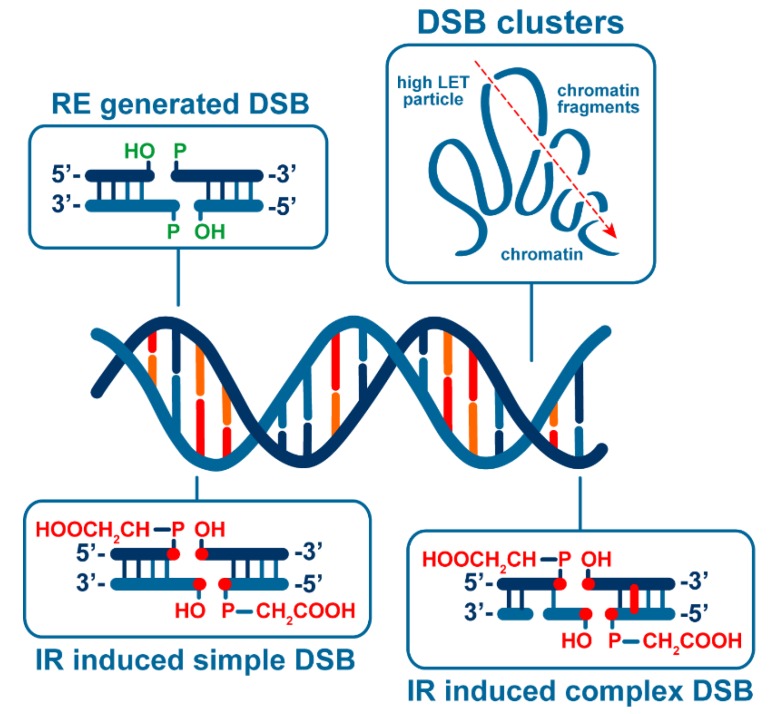
Concepts utilized in the classification of DSBs according to their degree of complexity. REs-generated DSBs break the DNA without chemically altering its nucleotides and are therefore considered the simplest form of DSBs (upper left box). IR induces DSBs by chemically altering at minimum two nucleotides in opposite strands mainly from ionization events within an ionization cluster (lower left box). The chemical entities generated in this way at the ends vary and need to be removed during DSB processing. This DSB is therefore more “complex” than that generated by REs. Since an ionization cluster may contain more than two events, the DSBs generated may be accompanied by additional forms of DNA damages near the DSB in the form of either additional single strand breaks or base damages. This generates DSBs of higher “complexity” in the sense that they will need more extensive processing (lower right box). Occasionally, particularly when cells are exposed to high LET particles, multiple DSBs are generated in some proximity (upper right box). The chromatin fragments generated in this way may be lost and are likely therefore to destabilize chromatin and compromise DSB processing. We therefore define them here as an additional level of DSB complexity. The consequences of defined generation of DSB clusters in cells are discussed in detail in the text.

**Figure 6 cancers-11-01671-f006:**
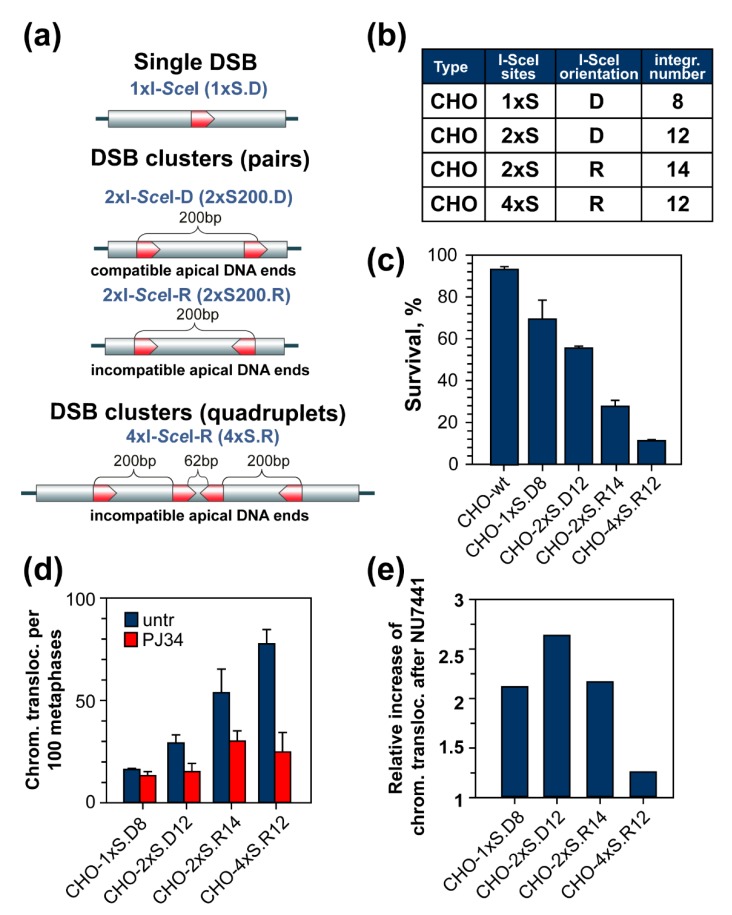
A model system to study the biological consequences of defined DSB clusters in Chinese Hamster Ovary (CHO) cells using engineered clusters of I-SceI recognition sequences. (**a**) Schematic representation of I-SceI constructs containing single DSBs, DSB pairs, and DSB quadruplets and integrated multiple times in the genome of CHO cells. (**b**) Description of the cell lines generated explaining the nomenclature used. The number of I-SceI sites (1×, 2×, 4×), their orientation (direct-D, reverse-R), and the number of integrations is presented in the table. (**c**) Clonogenic survival experiments of CHO cells harboring DSB clusters of different complexity. (**d**) Translocations forming in the same cell lines when analyzed at metaphase without any treatment, or after treatment with the PARP-1 inhibitor, PJ34. (**e**) Relative increase in chromosomal translocations after inhibition of c-NHEJ by incubating with the DNA-PKcs inhibitor NU7441. The results shown have been published and are redrawn here to illustrate the concepts developed in the review [57].

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
