# Peer review of "Necessities in the Processing of DNA Double Strand Breaks and Their Effects on Genomic Instability and Cancer"

_cancers, 2019, doi:10.3390/cancers11111671_

Round 1
Reviewer 1 Report
Dear authors,
I thank you for your interesting article. Hereby my comments.
Line 308: Please explain different ways of IR induce damage (direct and indirect effect). As heavy charged particles such as alpha have a greater probability of causing direct damage than low charged particles such as photon which causes most of its damage by indirect effect.
Line 419: Recently published “Cancers” an article from your own University Hospital, department of neuroanatomy = Dunker et al. Implementation of chick chorioallantonic membrane (CAM) model in radiation biology and experimental radiation oncology research. Please explain why this model was not suitable to test your hypothesis.
Author Response
Reviewer #1
General Comments
“I thank you for your interesting article. Hereby my comments.”
We thank the reviewer for the positive feedback. We hope that our study provides a useful overview of our ideas concerning coordination of DSB repair pathways that may be helpful to those working in the field.
Major Comments
“Line 308: Please explain different ways of IR induce damage (direct and indirect effect). As heavy charged particles such as alpha have a greater probability of causing direct damage than low charged particles such as photon which causes most of its damage by indirect effect.”We appreciate the suggestion of the reviewer. We provide an explanation of direct and indirect effects in the legend of Figure 1.
“Line 419: Recently published “Cancers” an article from your own University Hospital, department of neuroanatomy = Dunker et al. Implementation of chick chorioallantonic membrane (CAM) model in radiation biology and experimental radiation oncology research. Please explain why this model was not suitable to test your hypothesis.”The excellent overview article the Reviewer indicates focuses on the use of the Chick Chorioallantonic Membrane (CAM) Model in Radiation Biology and Experimental Radiation Oncology Research. The authors propose its implementation as a fast, cost-efficient model for semi high-throughput preclinical in vivo screening of the effects by molecularly targeted drugs. Our Review, on the other hand, focuses on events at the DNA, generated as a consequence of the induction of DSBs; it does not consider and cannot benefit, as the Reviewer points out, from the analysis of in vivo effects or the tissue models outlined in the recently published paper.

Reviewer 2 Report
This review article gives a nice summary of the different DNA repair pathways available to process DNA double strand breaks (DSBs), pointing out that ‘necessities’ rather than ‘choices’ underlie pathway engagement. These necessities are due to several factors, including chromatin destabilization and the degree of DSB complexity, as for instance caused by high LET radiation. Additionally, the authors point out the relation between the different pathways and the consequence in terms of genome integrity, ultimately speculating on their evolutionary role.
Major comments:
Figure 2, consider adding the cell cycle dependency under each pathway. Figure 3 and 4, Are these representative drawings? Or these graphs taken from actual data? If so please add the relative references adapted from [] etc. Also, error bars and data points are missing. Figure 3, it would be helpful to add data from a cell line deficient in HR and compare the survival after low and high LET radiation. There is evidence that deficiency in HR renders cells more sensitive to protons versus photons, although protons are actually considered low LET. Can the authours include a paragraph regarding protons? And perhaps also speculate on this in terms of stratification of patients carrying HR-related mutations (e.g. BRCA1/2) or use of PARP inhibitors. Can the authours add some additional information of the DNA fragments caused by low versus high LET radiation? This is something of increased importance for the immune response and therefore more data on this would be of interest for a larger audience. Page 9, line 335. It is stated that different LETs produce a similar number of DSBs. Can this be due to the inability of discriminating multiple DSBs within a cluster? Figure 6d and e, error bars are missing. Figure 6c and d, could you also include the cell survival data with the PARPi.
Minor comments:
Page 7, lines 278-280, confusing sentence please consider re-phrasing. Figure 3 and 4, could you add the LET of Fe ions.
Author Response
Reviewer #2
General Comments
“This review article gives a nice summary of the different DNA repair pathways available to process DNA double strand breaks (DSBs), pointing out that ‘necessities’ rather than ‘choices’ underlie pathway engagement. These necessities are due to several factors, including chromatin destabilization and the degree of DSB complexity, as for instance caused by high LET radiation. Additionally, the authors point out the relation between the different pathways and the consequence in terms of genome integrity, ultimately speculating on their evolutionary role”
We thank the reviewer for the positive evaluation of our paper.
Major Comments
“Figure 2, consider adding the cell cycle dependency under each pathway.”We thank the Reviewer for pointing out this omission. In the revised version of the manuscript we added panel b in Figure 2 showing the cell cycle dependency of the different DSB processing pathways.
“Figure 3 and 4, Are these representative drawings? Or these graphs taken from actual data? If so please add the relative references adapted from [] etc. Also, error bars and data points are missing.”The data shown in Figures 3 and 4 are extracted from earlier publications. We indicate this more clearly in the revised manuscript. We also added in the graphs the actual data points as requested by the Reviewer. To avoid cluttering in the manuscript, we refrained from adding error bars. The interested reader can always go back to the original paper for details along these lines.
“Figure 3, it would be helpful to add data from a cell line deficient in HR and compare the survival after low and high LET radiation.”We agree with the suggestion of the Reviewer and show as an example of homologous recombination deficient cell line in Figure 3b, results obtained with the irs1 mutant of V79 cells after exposure to X-rays and alpha particles. This mutant is defective in HRR as a consequence of defects in XRCC2 protein. The idealized curves were traced from Hill et al Radiat Res (2004) 162, 667-676, reference [62] in the revised version of the manuscript. The corresponding sentence is added on page 11, lines 4111-413.
“There is evidence that deficiency in HR renders cells more sensitive to protons versus photons, although protons are actually considered low LET. Can the authors include a paragraph regarding protons? And perhaps also speculate on this in terms of stratification of patients carrying HR-related mutations (e.g. BRCA1/2) or use of PARP inhibitors.”Again, we would like to thank the Reviewer for this relevant suggestion that also helps to emphasize the relevance of our Review in the therapy of cancer. The relevant paragraph with relevant references can be found on page 11, lines 442-448 in the revised manuscript.
“Can the authors add some additional information of the DNA fragments caused by low versus high LET radiation? This is something of increased importance for the immune response and therefore more data on this would be of interest for a larger audience.”The role of radiation-induced DNA fragments in the initiation and development of innate immune response is a hot topic at the present time and highly relevant for several aspects of cancer therapy. However, we found it very difficult to elaborate adequately on this topic without losing focus in a Review article that actually focus on different aspects of the radiation response. We hope the Reviewer will understand this confounding problem.
“Page 9, line 335. It is stated that different LETs produce a similar number of DSBs. Can this be due to the inability of discriminating multiple DSBs within a cluster?”This is indeed possible and it is now mentioned on page 9, lines 341-343.
“Figure 6d and e, error bars are missing. Figure 6c and d, could you also include the cell survival data with the PARPi.”We added error bars in Figures 6c and d. Figure 6e shows ratios generated from results similar to those shown in Figures 6d and have therefore no error bars. We do not have results with Parp1 inhibitors to add in Figure 6c. However, we plan to generate such data in the near future.
Minor Comments
“Page 7, lines 278-280, confusing sentence please consider re-phrasing..”We rephrased the passage.
“Figure 3 and 4, could you add the LET of Fe ions..”The LET value for the 56Fe ions is 151 keV/µm. This is now indicated in Figures 3 and 4.
